# An e-Problem-Based Learning Program for Infection Control in Nursing Homes: A Quasi-Experimental Study

**DOI:** 10.3390/ijerph192013371

**Published:** 2022-10-16

**Authors:** Young-Rim Choi, Ye-Na Lee, Dayeong Kim, Won Hee Park, Dai Young Kwon, Sung Ok Chang

**Affiliations:** 1College of Nursing, Korea University, Seoul 02841, Korea; 2Department of Nursing, University of Suwon, Hwaseong 18323, Korea; 3College of Nursing, BK21 FOUR R&E Center for Learning Health Systems, Korea University, Seoul 02841, Korea; 4Gifted Education Center, Korea University, Seoul 02841, Korea

**Keywords:** nursing homes, infection, e-PBL, critical thinking, nursing staff, education

## Abstract

Infection is a significant factor adversely affecting the health of nursing home (NH) residents, potentially even leading to death. Therefore, educating NH staff to think critically is necessary to prevent and control infection. In this study, we developed an electronic problem-based learning (e-PBL) program using the Network-Based Instructional System Design model to enhance South Korean NH staff’s critical thinking competencies; subsequently, its effectiveness was evaluated. This study utilized a quasi-experimental nonequivalent pretest–post-test design. The participants (n = 54) were randomly allocated into an experimental group (n = 28) and a control group (n = 26). The results indicate that the e-PBL program significantly improved the critical thinking disposition in the experimental group compared with in the control group.

## 1. Introduction

Frail residents in high-density nursing homes (NHs) are at high risk of infectious diseases [1,2]. Such residents are vulnerable to various infections, including urinary tract and skin infections and respiratory infections such as COVID-19 [3]. Such infections adversely affect the health of NH residents and are significant risk factors that can even lead to death [4]. Approximately two million infection cases occur annually in NHs in the United States, increasing mortality [5], making infection control paramount in NH settings. The Factors influencing the incidence of infection are reportedly related not only to the residents’ frailty and underlying diseases but also to NHs’ staff lack of knowledge and training [6,7]. Therefore, providing an effective intervention to NH nursing staff who provide direct care could improve infection control competences, ultimately resulting in positive outcomes for residents’ health. NH nursing staff provide direct care to NH residents; therefore, their infection control competence is inherently related to residents’ health [8]. 

In NH settings, the early detection and prevention of infections is paramount, while in acute medical settings the focus lies on treatment [9]. Recognizing the risk factors of infection that may have a significant impact on residents’ health and establishing strategies to prevent them in advance should be a thinking process of logical ways to bring about changes for the improvement [7,10].

Critical thinking refers to an individual’s competence in applying higher-order cognitive skills and his/her disposition to deliberate on a logical course of action for solving new problems [11]. Chang et al. have demonstrated that critical thinking ability is significantly associated with nursing competence [12]. To recognize the risk factors of infection that may have a significant impact on residents’ health and to establish strategies to prevent them in advance, nursing staff should critically perceive any new problems concerning residents and think of logical ways to improve infection prevention [10,13]. In addition, accurate and grounded thinking processes and practical competence are important to critically recognize a resident’s physical changes based on the knowledge of the clinical symptoms of infection and respond quickly to them [13]. Therefore, NH staff’s critical thinking in devising and implementing strategies for the early detection and prevention of infectious diseases is crucial.

PBL is known as an effective educational strategy that can improve critical thinking in many healthcare disciplines [14,15]. PBL-based education promotes questioning the given phenomenon or knowledge in the quest for creative solutions to problems [15]. Although many efforts have been made to utilize PBL education in healthcare, they faced limitations, such as securing a space for implementing education and learners finding the necessary time fairly lengthy [16]. In response to this, electronic PBL (e-PBL), an educational method that combines web-based learning and PBL, has emerged as an alternative. In e-PBL, students learn in a web environment free from time and space constraints and can also be an answer to situations such as the current COVID-19 pandemic [14,17]. In this study, we developed an e-PBL-based infection control education program for NH staff to improve critical thinking disposition in preventing and addressing infection.

## 2. Background

PBL is a new learning paradigm that emerged in the mid-1960s [18]. It was developed based on constructivist epistemology. In existing objectivist education theory, the teacher became a supervisor and the learner the passive receiver. However, in constructivist teaching theory, the learner is an active being, and the teacher is a facilitator who plays the role of helping the learner [19]. PBL is a self-directed learning process combining cooperative learning and self-directed learning in which the learner becomes the subject and solves problems [19].

The teacher develops problems based on unstructured situations and presents them as assignments to students. The teacher also guides students in collecting and exploring learning materials to help them solve these tasks. The PBL process structured in eight steps are as follows: (1) arousing students’ doubts about problems, (2) prior to studying problems, (3) raising questions, (4) defining the scope of knowledge, (5) proposing an information acquisition plan, (6) conducting research, (7) sharing information and (8) concluding [20]. In this process, students can acquire problem-solving skills through critical thinking [21].

## 3. Methods

This study’s research process was divided into the educational program’s development and the evaluation of its effects.

### 3.1. Developmental Process of the Educational Program

This study followed the Network-Based Instructional System Design model to develop the educational program [22]. The model comprises five steps: analysis, design, production, implementation, and evaluation (Figure 1).

#### 3.1.1. Step 1. Analysis

In Step 1, the characteristics of the learners and the educational contents were identified by analyzing learners’ needs and conducting a literature review. To analyze learners’ needs, in-depth interviews were conducted with nine practitioners in NHs. The average age of the participants was 48.25 ± 11.76 years. All of the participants had used the e-learning training program before. Each individual’s work experience was varied, with an average of 5.44 ± 2.50 years. Eight registered nurses and one nurse assistant participated in the interviews, which lasted 60 min. Semi-structured questions regarding infection control education goals and contents, educational media, and training time were used. The findings of the in-depth interview are as follows. The goals and contents in NHs focus on early detection and prevention rather than on the treatment of infections. Furthermore, e-learning has been confirmed as a suitable educational medium due to the nature of the intervention during the COVID-19 pandemic and the nursing staff’s workloads. Regarding training time, it was decided that 2–3 h was an appropriate duration. The literature review showed that PBL is a suitable educational method for improving nurses’ judgment for early detection and prevention of infection [23]. The practical reasoning process in recognizing and dealing with new infectious cases could be improved by an educational strategy focused on problem-solving [13]. Moreover, as e-learning improves learning ability through self-directed learning and repetition [24], the program’s educational effect was enhanced through e-PBL, which combines the advantages of PBL and e-learning [25]. The e-learning technology and environment necessary for this study were considered during the program’s development process through discussions between two computer education experts and two doctors in nursing.

#### 3.1.2. Step 2. Design

The program was designed considering the educational goals and contents identified in the analysis stage, the educational medium, and the training time. Text and multimedia materials suitable for the learning topic were collected and composed in PowerPoint. The PBL contents were constructed based on various infection control cases collected from the in-depth interviews to enhance the suitability of the problem scenario. For communication between learners and educators, technical support was established via instant messaging, chatbot.

#### 3.1.3. Step 3. Production

The first author (Y-R.) created a storyboard based on the educational content confirmed at the design stage and then discussed it with the e-learning program producer to develop the program. The program contents were finalized by modifying and supplementing the educational contents through discussion between one professor of nursing and two doctors of nursing. The program was developed to be optimized in the Chrome environment as a software tool. The expanded educational program was pilot-tested over two weeks.

#### 3.1.4. Step 4. Implementation

Participants joined guided meetings in which they were trained on how to use the program and the related procedures, and the necessary guide materials were provided. The education program was made available to NH staff through a website for two weeks, beginning 30 December 2021. The research team responded to the learners’ questions regarding the use of the program and learning contents through a chatbot and checked the web page for errors.

#### 3.1.5. Step 5. Evaluation

The critical thinking of NH staff was evaluated to judge the effectiveness of the e-PBL education program. Granting that the PBL education improves learner’s critical thinking abilities in clarifying, understanding, and analyzing problems and making inference for creative solutions, evaluating critical thinking means measuring a long-term meaningful improvement of education [26,27]. The better the critical thinking ability of nursing staff is, the better the individual’s ability to manage infection prevention and control [28]. Therefore, the Critical Thinking Disposition Tool developed by Yoon [29] was employed before and after the educational program.

### 3.2. Effects of the Educational Program

#### 3.2.1. Design

A nonequivalent control group pretest–post-test design was used (Figure 2). We compared the effects among South Korean NH staff who had taken part in the e-PBL-based educational program and those who were not exposed to the program. The control group was exposed to e-learning without PBL strategy to prevent any possible threats to the internal validity of the experiment and to identify the PBL-specific outcome effects.

#### 3.2.2. Participants

This study was conducted with 54 NH staff working in Seoul and Gyeonggi Province in South Korea. The selection criteria were as follows: currently working and providing direct nursing care as nurses, care workers, or nursing assistants in NHs. To assess the effects of the education program, individuals who had difficulties using computers were excluded from this study. Using G*Power 3.1.9.4 software, it was determined that for an effect size of 0.80, significance level of 0.05, and power of 0.80, a sample size of 52 would be appropriate [30]. A total of 59 participants were recruited and consented voluntarily to be part of the study. Those who did not complete the training program or did not respond to the survey were excluded. Finally, 54 participants completed the training program—28 participants in the experimental group and 26 in the control group. 

#### 3.2.3. Outcome Measures 

The Critical Thinking Disposition Tool developed by Yoon was used to assess the participants’ critical thinking [29]. The scale comprises 27 items across 7 subscales: intellectual eagerness/curiosity (5 items), prudence (4 items), self-confidence (4 items), systematicity (3 items), intellectual fairness (4 items), healthy skepticism (4 items), and objectivity (3 items). Each item is scored on a 5-point Likert scale ranging from 1 to 5, where 1 = “absolutely do not agree” and 5 = “absolutely agree”. Higher scores indicate a more developed critical thinking disposition. In Yoon [29]’s study, the instrument’s reliability was Cronbach’s α = 0.85, while in this study it was 0.89. Each of the aforementioned subscales are defined below.

Intellectual eagerness/curiosity: Having a desire to learn and an attitude to search for answers to problems and to pose new problems.Prudence: Withholding judgment and pursuing results accurately until valid and sufficient evidence is obtained.Self-confidence: Believing in one’s reasoning process and drawing valid conclusions through critical thinking.Systematicity: Not deviating from the core of the problem being discussed, exploring it systematically and in an orderly manner, and maintaining logical consistency until reaching a conclusion.Intellectual fairness: Fairly evaluating one’s own and others’ opinions while seeking the truth with an open mind.Healthy skepticism: Simultaneously acknowledging and questioning generally accepted facts.Objectivity: Drawing conclusions based on valid evidence and taking a position when evidence and reasons are sufficient.To measure the participants’ satisfaction with the program contents, we used a 5-point Likert scale (1 = absolutely not satisfied to 5 = absolutely satisfied).

### 3.3. Data Analysis

IBM SPSS version 25.0 software was used to analyze the data. Descriptive statistics and a two-sample *t*-test were performed to confirm respondents’ differences in critical thinking disposition before and after the program. All analyses were validated at a significance level of 0.05.

### 3.4. Ethical Considerations

Ethical approval was obtained from the institutional review board of the first author’s university (KUIRB-2021-0057-01). Written informed consent was obtained from all participants.

## 4. Results

### 4.1. Development of the e-PBL-Based Infection Control Educational Program for Nursing Home Staff

The e-PBL-based infection control education program developed in this study was designed to solve problematic situations related to infection control in NH settings through a step-by-step learning process. Table 1 presents the curriculum of the educational program. The educational program consists of three chapters and aims to teach NH staff how to properly manage and identify signs and symptoms of infection in NH settings. 

Figure 3 shows the educational program’s slides depicting the program’s introduction, as well as information on case-based learning, multimedia learning, PBL, using the chatbot, etc.

### 4.2. Effects of the e-PBL-Based Infection Control Educational Program for Nursing Home Staff

Table 2 shows the general characteristics of the participants of this study. There were no statistically significant demographic differences between the experimental and control groups.

There was a statistically significant difference between the two groups, with a difference of 0.15 ± 0.14 points in the pre- and post-scores for the 27 questions in the critical thinking disposition of the experimental group and 0.04 ± 0.07 in the control group (t = −5.006, *p* < 0.001). However, among the subcategories of healthy skepticism (t = −0.542, *p* = 0.590), intellectual eagerness/curiosity (t = −0.373, *p* = 0.710), and intellectual fairness (t = 1.039, *p* = 0.304), there was no statistically significant difference between the two groups. However, for self-confidence (t = −4.50, *p* = <0.001), systematicity (t = −4.011, *p* = <0.001) prudence (t = −2.242, *p* = 0.029), and objectivity (t = −2.512, *p* = 0.015), we identified a statistically significant difference between the two groups (Table 3).

## 5. Discussion

In high-density NHs, where residents are vulnerable to various infectious diseases, including COVID-19, it is crucial to improve the staff’s competency to prevent and care for them. This study developed and evaluated an educational program for improving the critical thinking disposition of NH staff to improve their infection control capabilities.

### 5.1. Development the Educational Program

The key contribution of this study is the development of an effective educational program to improve the critical thinking disposition of NH staff for the early detection of infection signs and symptoms of NH residents. 

The goal of the educational program developed in this study was to improve NH staff’s capability for the early detection of infection signs and symptoms in NH residents. Early detection is the most important factor for resident care in NHs [31,32] because these are residential facilities for individuals who require 24 h assistance due to physical and/or cognitive impairments [33]. This is in contrast to acute care settings, where the focus lies on treating patients’ diseases [32]. The educational content was structured in stages, from the definition of infection to the identification of its signs and symptoms. Participants expressed their satisfaction with the educational content of the program with a score of 3.5 out of 5, suggesting that the program meets the needs of NH staff, although has room for improvement. The educational goals and contents of the program were developed through the integration of theoretical (literature review) and practical methods (interviews). 

Web-based education has become commonplace in various educational fields, especially after the COVID-19 pandemic [34,35]. Furthermore, e-learning has been shown to better improve individuals’ knowledge and skills compared with traditional learning [36]. Although e-learning has a weakness, in that it makes learners passive participants through lecture-dominant teaching, PBL is an active educational method that is learner-centered during the learning process, and it promotes critical thinking and self-directed learning [37,38,39]. In particular, e-PBL, which combines web-based instruction and PBL, provides a self-directed learning environment for learners to actively solve problems without time and space constraints. This educational method can foster reflective attitudes and critical thinking [17]. The difference between the web-based education of the educational program developed in this study is that learner–learner interactions (e.g., sharing ideas and recommendations) and teacher–learner interactions are made possible via chatbots. However, to enhance the program’s accessibility, it will be necessary to adapt it to a mobile device environment.

### 5.2. Effects of the Educational Program

The findings confirmed that the critical thinking of NH staff improved significantly after using the e-PBL-based education program. In the existing literature, PBL has been widely used as an educational method to stimulate critical thinking; however, prior studies’ findings are inconsistent. Some studies have shown that PBL improves critical thinking [15,40], while others have not [41]. Nevertheless, according to a recent systematic review on PBL and critical thinking, PBL can better support the development of critical thinking when compared with traditional methods [42]. 

In this study, an e-PBL-based educational program significantly improved the total scores measuring the critical thinking disposition. This indicates that the education program is effective at improving the NH staff’s critical thinking involved in incorporating their knowledge and residents’ sign and symptoms of infection to determine possible solutions [11,27]. However, there was no significant difference between the experimental and control groups in the healthy skepticism and intellectual fairness categories of critical thinking disposition. This was likely due to the relatively short training that participants underwent. Given nursing staff’s work overload due to the COVID-19 pandemic [43], long-term research in NHs is difficult; therefore, our research team had to develop a program for maximum educational effect in the shortest amount of time. In a future study with a more extensive training period, an explicit comparison will be possible. 

Both groups exhibited no statistically significant difference in intellectual eagerness/curiosity. This is because extroverted learners have been reported to have high intellectual eagerness/curiosity and introverted learners to possess high prudence [30]. Although it was not statistically significant, the intellectual eagerness/curiosity of the experimental group improved after the training; thus, this factor could potentially be improved through education. Systematization and self-confidence significantly improved in the experimental group compared with the control group, which demonstrates the effectiveness of the PBL education method. PBL is a step-by-step process from problem to solution. We believe that this characteristic of the program improved learners’ systematicity and self-confidence. To date, little research on PBL-based education for NH staff has been conducted, but PBL-based education methods may be suitable for educating nursing practitioners in NHs [44]. In addition, PBL fosters self-confidence among learners, enhancing the belief that their reasoning process will lead them to a valid conclusion.

### 5.3. Limitations

One limitation of this study was that the participants interviewed were a convenience sample; however, all participants were qualified staff in NHs with more than 5 years of working experience who could practice infection control for residents. Furthermore, we could not measure changes in critical thinking disposition over time. It is therefore necessary to confirm the program’s long-term effects via studies with longer durations.

The purpose of this study was to suggest an effective way to improve infection management by enhancing the critical thinking and problem-solving ability of NH staff rather than to immediately reduce the infection rate; therefore, we did not measure the exact change in the incidence of residents’ infections. It will be possible to conduct a study that targets a reduction in the incidence of infection in NHs over a lengthy period of time through other research designs in the future.

### 5.4. Implications for Nursing Practice and Education

The developed e-PBL-based educational program sheds light on the new direction of NH education. A chatbot-assisted interactive web program can be employed to help NH staff improve their critical thinking disposition to solve infectious problems. Unlike conventional education, which is constrained by time and space, this online education could generate more effective outcomes and increase learners’ satisfaction. The program could be adapted to include nursing problems other than infection control, expanding on the ways NH staff could by instructed in competent practices for resident care.

## 6. Conclusions

Pandemics such as COVID-19 pose a considerable risk to NH residents’ health. This study identified the effect of an e-PBL-based education program for infectious disease management competency on NH staff’s critical thinking disposition. Based on the present findings, we propose the development of a study on the long-term effects of the program and the development of mobile device education content.

## Figures and Tables

**Figure 1 ijerph-19-13371-f001:**
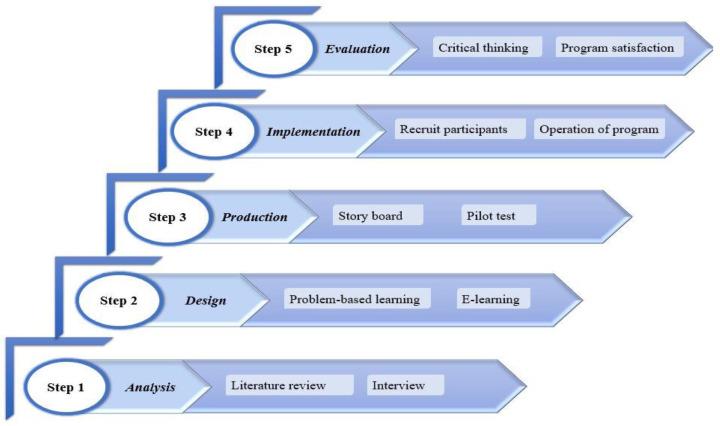
Developmental process of the educational program.

**Figure 2 ijerph-19-13371-f002:**
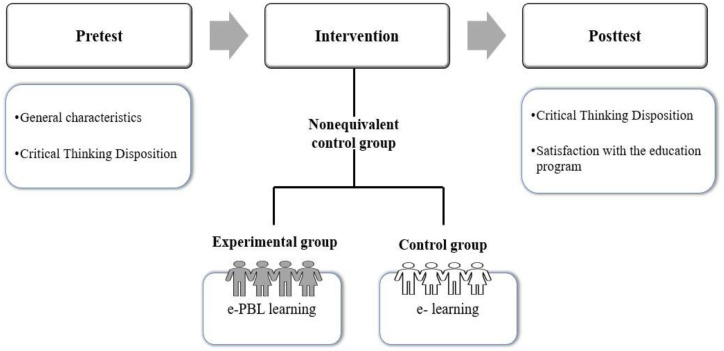
Experimental design.

**Figure 3 ijerph-19-13371-f003:**
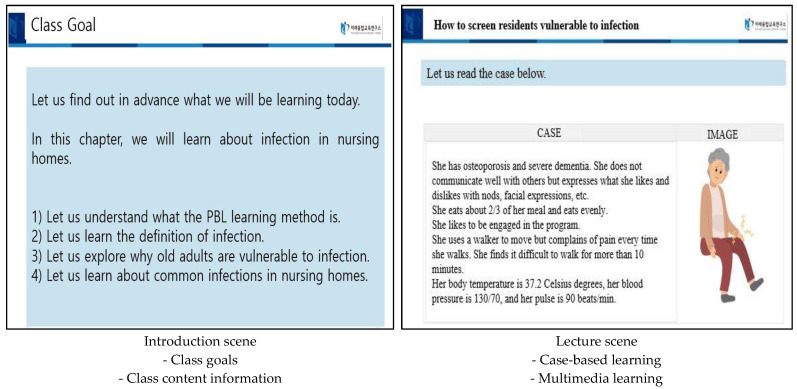
Example slides of the e-PBL educational program.

**Table 1 ijerph-19-13371-t001:** Course for the infection control educational program for NH staff.

Chapter	Title	Curriculum	e-PBL based Activities
1	Common infections in NHs	- Definition of infection - Reasons why older people are vulnerable to infections - Common types of infections in nursing homes	- Watching video footage - Identifying the problem
2	Signs and symptoms of infection	- Transmission routes of infection - Definition of signs and symptoms - Typical and atypical signs and symptoms of infection in nursing home residents	- Performing individual and group assignments - Discussion with other learners to solve the problem
3	Identify signs and symptoms of infection	- Screening residents who are vulnerable to infection - Assess and manage signs and symptoms of infection	
4	Review	- Review each chapter	- Self-evaluation - Peer review about the activities

**Table 2 ijerph-19-13371-t002:** General characteristics of the participants (N = 54).

Variable	Category	N (%) or M ± SD	
		Exp. (N = 28)	Cont. (N = 26)
Age (years)	Average	56.64. ± 8.72	55.15 ± 7.09
Gender	Male	1(3.6)	0
	Female	27(96.4)	26(100)
Education level	High-school	12(42.8)	12(46.2)
	Associate degree	6(21.4)	9(34.6)
	Bachelor’s degree	5(17.9)	4(15.4)
	≥Master’s degree	5(17.9)	1(3.8)
Job	Registered nurse	14(50)	11(42.3)
	Nurse assistant	3(10.7)	6(23.1)
	Care worker	11(39.3)	9(34.6)
Total work experience (years)		5.81 ± 4.25	5.42 ± 2.74

Note. Exp. = experimental group; Cont. = control group; M = mean; SD = standard deviation.

**Table 3 ijerph-19-13371-t003:** Effects of the e-PBL-based infection control educational program for NH staff (N = 54).

Category of Items (Range 1–5)	Time	Mean ± SD		t	*p*
		Exp. (N = 28)	Cont. (N = 26)		
Healthy skepticism	Pretest	3.34 ± 0.69	3.31 ± 0.71	−0.250	0.804
	Post-test	3.38 ± 0.69	3.32 ± 0.72	−0.399	0.692
	Difference	0.57 ± 0.21	0.03 ± 0.08	−0.542	0.590
Intellectual fairness	Pretest	3.28 ± 0.73	3.34 ± 0.80	0.159	0.874
	Post-test	3.31 ± 0.74	3.35 ± 0.81	0.420	0.677
	Difference	0.00 ± 0.18	0.05 ± 0.16	1.039	0.304
Objectivity	Pretest	3.36 ± 0.79	3.37 ± 0.97	0.061	0.952
	Post-test	3.35 ± 0.79	3.33 ± 0.88	−0.802	0.426
	Difference	0.14 ± 0.31	0.04 ± 0.22	−2.512	0.015 *
Systematicity	Pretest	2.99 ± 0.59	2.83 ± 0.59	0.943	0.350
	Post-test	3.29 ± 0.48	2.99 ± 0.60	−2.101	0.040 *
	Difference	0.46 ± 0.59	0.00 ± 0.14	−4.011	<0.001 *
Prudence	Pretest	3.43 ± 0.73	3.40 ± 0.78	−0.167	0.868
	Post-test	3.51 ± 0.65	3.37 ± 0.73	−0.762	0.450
	Difference	0.89 ± 0.22	−0.02 ± 0.12	−2.242	0.029 *
Intellectual eagerness/ Curiosity	Pretest	3.55 ± 0.81	3.52 ± 0.83	−0.121	0.904
	Post-test	3.58 ± 0.76	3.53 ± 0.77	−0.226	0.822
	Difference	0.36 ± 0.11	0.02 ± 0.26	−0.373	0.710
Self-confidence	Pretest	2.97 ± 0.60	3.01 ± 0.60	−0.106	0.916
	Post-test	3.29 ± 0.47	2.94 ± 0.56	−2.50	0.016 *
	Difference	0.27 ±0.35	−0.07 ± 0.18	−4.50	<0.001 *
Total	Pretest	3.27 ± 0.56	3.28 ± 0.65	0.066	0.948
	Post-test	3.42 ± 0.47	3.28 ± 0.61	−0.977	0.333
	Difference	0.15 ± 0.14	−0.04 ± 0.07	−5.006	<0.001 *

Note. Exp. = experimental group; Cont. = control group; M = mean; SD = standard deviation. * = *p* < 0.05.

## Data Availability

Data that support the findings of the study are available upon reasonable request from the corresponding author.

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
