# Peer review of "An e-Problem-Based Learning Program for Infection Control in Nursing Homes: A Quasi-Experimental Study"

_ijerph, 2022, doi:10.3390/ijerph192013371_

Round 1
Reviewer 1 Report
Authors present their e-PBL program to enhance critical thinking competences in nursing home staff. The article is well structured, for the development of the educational program it uses Network-Based Instructional System Design. The process of design, implementation and evaluation is described in a thoughtful and illustrative way.
There are some formal minor errors, for example in line spacing, see line 30-31. The numbering and line spacing in the References needs a thorough revision, see 1. 1. Kojima, G. Prevalence of Frailty in Nursing Homes:..
Reviewer 2 Report
Dear Authors:
Thank you for sharing your work throughout this manuscript.Here my observations:
1. The format must be corrected, there are lines that have been moved, for example, on lines 30, 34, 45 and 46. In the bibliography there is double numbering, I presume that some lines have been moved.
2. In the introduction, you wrote about the risks of infection of patients in nursing homes and establish that the lack of knowledge and training would be one of the factors that influence the incidence of infections. To hold this sentence you used only one reference (6) and it would be good for the reader if this sentence had much more evidence.
3. In the third paragraph you mentioned the critical thinking disposition score. I think it would be good for the article if you could tell us about this measure and why it is used to evaluate the success of a problem-based learning program, and what other measures exist that you considered but discarded for some reason. In short, if this score is a gold standard.
4. Along these same lines, you mentioned that problem-based learning is an effective tool to increase critical thinking. Again, it would be very useful to support this claim in more references than just one (12).
5. It would be recommended for the article to know why they decided on this educational alternative compared to other possible options.
6. Although the introduction begins by contextualizing the problem of infections in nursing home residents, the measurement of the intervention is the critical thinking disposition score. Then, the educational topic could have been any other, unless the measurement tool was adapted to the topic, which is not described in the article.
7. A major limitation of the study is that it fails to review the real impact of the intervention on the incidence of patient infections. I understand that it was not the focus of the study, and that this would be with a different research method and over a longer period of time.
Reviewer 3 Report
Thanks for giving me the opportunity to review this manuscript. Overall, it is a well-written, interesting, and organized paper. I have the following comments/suggestions:
1. The authors mention stats about the cases of infection in the United States. I recommend adding stats about nursing homes in South Korea, if available.
2. On page 2 (lines 74-75), the authors mentioned that the duration of training time was determined as 2-3 hours. This has to be supported by the findings of the in-depth interviews.
3. I suggest adding more details regarding the expertise of the personnel involved in the development of the educational program. For example, the total experience of each individual and previous work in program development.
4. Regarding the sample size estimation (page 4), what was the effect size used to determine the number of participants. Also, the authors stated that the level of significance was set at .50. They probably means .05.
5. In the results section, the authors reported the results of the t-tests without reporting the ANOVA results. However, they mentioned (in the data analysis subsection) that repeated measures ANOVA was used in this study.
6. Participants' satisfaction with the education program was listed on Figure 2. However, there was no mention of how it was measured. The results regarding the satisfaction with the program were not presented in the results section either.
7. I suggest adding a subsection in the discussion to discuss the implications of the current study to nursing practice and education.
8. Fix the numbering of the references in the reference list. There is a duplication of numbering. For example, the first reference is numbered (1) twice.
Good luck!
Reviewer 4 Report
Thank you for the possibility to read this exciting manuscript. I found the topic and contexts of the study critical but unfortunately, the conduction and theoretical background are insufficiently described—also, the reason why this study has been conducted should have been elaborated more broadly.
The biggest flaw of the manuscript is in the theoretical/conceptual background. The key concepts such as BPL (e-BPL) and critical thinking and related research are poorly presented. Why has this specific e-BPL been selected and why are these measures? Why was the training developed as it is? The authors have described their method including a literature review as a first step. it would be interesting to know more about the "findings" of this review because they seem to include the reason for the whole conduction of this study. What is the aim of this study in the first place?
Why control group is e-learning group, but e.g. a group without any training? What is the real added value of the findings (something that we did not already know)? Discussion should include broader theoretical discussion and implications as well as critical methodological discussion.
Round 2
Reviewer 2 Report
Dear authors:
Thanks to accept the observations.
Take care in line 318, item 6 conclusions was written in the previous paragraph.
Kind regards
Author Response
We appreciate the very helpful comments on the first revision. The manuscript has been strengthened significantly with your guidance.
We revised the format error as your advice (Page 11, Line 326). Thank you very much for your comments.
Reviewer 3 Report
Thanks for your detailed response. The authors have sufficiently addressed my comments.
Author Response
We appreciate the very helpful comments on the first revision. The manuscript has been strengthened significantly with your guidance.
Thank you much for your positive comments.
Reviewer 4 Report
The revised version looks good and the current performance may be accepted. However, if you still have some energy and the editor so decides, the description of the concept and method of PBL might still be strengthened. I would also like to see a stronger judgment of why PBL has been utilized as a method here from hundreds of other educational and training possibilities...
